# Understanding COVID-19 Vaccine Acceptance among Healthcare Workers in Indonesia: Lessons from Multi-Site Survey

**DOI:** 10.3390/vaccines12060654

**Published:** 2024-06-12

**Authors:** Madan Khatiwada, Ryan Rachmad Nugraha, Carine Dochez, Harapan Harapan, Kuswandewi Mutyara, Laili Rahayuwati, Maimun Syukri, Eustachius Hagni Wardoyo, Dewi Suryani, Bertha J. Que, Cissy Kartasasmita

**Affiliations:** 1Network for Education and Support in Immunisation (NESI), University of Antwerp, 2610 Antwerp, Belgium; 2USAID Health Financing Activity/ThinkWell, Jakarta 10110, Indonesia; 3Medical Research Unit, School of Medicine, Universitas Syiah Kuala, Banda Aceh 23111, Indonesia; 4Department of Microbiology, School of Medicine, Universitas Syiah Kuala, Banda Aceh 23111, Indonesia; 5Department of Public Health, Faculty of Medicine, Universitas Padjadjaran, Bandung 45363, Indonesia; 6Faculty of Nursing, Universitas Padjadjaran, Bandung 45363, Indonesia; 7Department of Internal Medicine, School of Medicine, Universitas Syiah Kuala, Banda Aceh 23111, Indonesia; 8Department of Microbiology, School of Medicine, Universitas Mataram, Mataram 83125, Indonesia; 9Faculty of Medicine, Universitas Pattimura, Ambon 97233, Indonesia; 10Department of Pediatric, Rumah Sakit Umum Pusat Hasan Sadikin, Bandung 40161, Indonesia

**Keywords:** COVID-19, COVID-19 vaccine, vaccine acceptance, healthcare workers, Indonesia

## Abstract

The COVID-19 pandemic presented an unprecedented challenge to public health as well as an extraordinary burden on health systems worldwide. COVID-19 vaccines were attributed as a key tool to control the pandemic, with healthcare workers (HCWs) as a priority group to receive the vaccine. Healthcare workers are considered one of the most trusted sources of information on vaccines and vaccination. This study was conducted to evaluate the acceptability of the COVID-19 vaccine among HCWs in four different provinces of Indonesia. An anonymous cross-sectional study was conducted online among HCWs between December 2020 and February 2021. Out of 2732 participants, 80.39% stated that they would accept the COVID-19 vaccine, while 19.61% were hesitant to receive the vaccine. Concerns about the safety profile of COVID-19 vaccines and potential side-effects after vaccination were the main reasons among the participants to refuse the vaccine. Male gender, single status, higher education level, and higher risk perception increased the acceptability of the COVID-19 vaccine. Other motivators of COVID-19 vaccine acceptance include a high level of trust in the government and increased confidence in vaccine safety and efficacy studies. Dissemination of information in a timely manner as well as training programs for HCWs are crucial to increasing confidence in the COVID-19 vaccination program.

## 1. Introduction

The Coronavirus Disease 2019 (COVID-19) pandemic resulted in a massive global death toll and posed an immense challenge to public health systems [1]. The disease, caused by the SARS-CoV-2 virus, was declared a Public Health Emergency of International Concern (PHEIC) by the World Health Organization (WHO) on 30 January 2020, and was later classified as a pandemic on 11 March 2020 [2]. Since the pandemic began, over 676 million cases and more than 6.9 million deaths related to COVID-19 have been reported globally as of 15 February 2024 [3]. Indonesia, in particular, was heavily impacted, with over 6.8 million cases and more than 161,000 deaths [4].

On 5 May 2023, after over three years of the pandemic, the WHO announced that COVID-19 was no longer a PHEIC due to a continuous decline in case numbers and deaths [2]. COVID-19 vaccines have been a crucial component of the preventative measures taken by various countries, significantly aiding in the control of the pandemic. A modeling study indicated that COVID-19 vaccinations prevented approximately 14.4 million deaths across 185 countries and territories between December 2020 and December 2021 [5]. By 15 February 2024, over 13.5 billion vaccine doses had been administered globally, with 70.6% of the world’s population receiving at least one dose of the COVID-19 vaccine [6].

Despite the need to vaccinate the majority of the population globally to achieve herd immunity, several factors negatively impacted the uptake of the COVID-19 vaccine [7,8,9]. Among many other factors, vaccine hesitancy was a hindering factor that delayed the uptake of COVID-19 vaccines [7,9]. The WHO defines vaccine hesitancy as a “delay in acceptance or refusal of vaccines despite availability of vaccination services” [10]. Other factors include lack of knowledge, accessibility issues, systematic and operational challenges, distribution problems, and cold chain readiness [11].

Even though vaccines are considered one of the most effective public health interventions, the increasing amount of misinformation and disinformation around vaccines and vaccination has negatively influenced people’s attitudes towards vaccines and vaccination in general, leading to low acceptance of the COVID-19 vaccine [12,13]. Particularly with the development of COVID-19 vaccines in a shorter timeframe compared to other traditional vaccines, many people were skeptical about receiving the vaccines against COVID-19 and in making decisions for their children to receive the vaccines [12]. The abundance of information available on social media channels about COVID-19 and COVID-19 vaccines has increasingly made it challenging to distinguish fact-based information from inaccurate, false information, further negatively impacting people’s decision to get vaccinated [14,15].

A high level of vaccine hesitancy undermines COVID-19 vaccination efforts, ultimately putting society and the community at greater risk of the disease, primarily healthcare workers (HCWs). Frontline HCWs are more susceptible to contracting COVID-19 [16]. Therefore, the WHO designated them as a top priority for receiving COVID-19 vaccines once they were available [17]. Vaccinating healthcare workers is crucial for lowering infection rates, preventing severe COVID-19 complications, and alleviating the pressure on health systems due to healthcare staff shortages. In addition, healthcare workers are considered one of the most trusted sources of information on vaccines and vaccination by the general public [18]. Therefore, negative sentiments towards the COVID-19 vaccine and vaccination among healthcare workers will profoundly influence the general public’s perception of the vaccines [19]. Several studies have demonstrated that vaccine acceptance among the general public declines if it is not recommended or if concerns are raised by their respective healthcare workers [19,20]. Several studies conducted in different countries have demonstrated varied levels of COVID-19 vaccine acceptance among healthcare workers [19,20,21,22]. It is of utmost importance to understand the factors that influence healthcare workers’ attitudes towards COVID-19 vaccines and vaccination to develop interventions and strategies to address the concerns and build confidence in vaccines and vaccination programs.

This study aims to investigate the acceptability of COVID-19 vaccines and identify potential factors that influence vaccine acceptance among healthcare workers in various provinces of Indonesia. The findings from this research were crucial in shaping COVID-19 vaccine implementation strategies and crafting effective communication approaches to boost vaccine uptake among healthcare workers in Indonesia.

## 2. Materials and Methods

### 2.1. Study Design and Development of the Questionnaire

This study was conducted during the first wave of COVID-19 vaccination in Indonesia (from 23 December 2020 to 15 February 2021). Healthcare workers included in the study were: (1) 18 years old or older, (2) worked in a hospital/clinic providing services to COVID-19 patients, and (3) worked in a functional unit or were responsible for patient care. The exclusion criteria were HCWs younger than 18 years and those working in hospitals not included in the study. The questionnaire was developed through a literature review of past research studies [23,24,25,26] and close consultation with the Indonesian health authorities.

The questionnaire was developed with the intent of capturing healthcare workers’ perceptions towards COVID-19, COVID-19 vaccines, and vaccination. Such a questionnaire entails an assessment of the sociodemographic profile, vaccination profile, and vaccine hesitancy profile.

Sociodemographic profile: Sociodemographic variables assessed include gender, age, marital status, religion, region, educational level, job title, health insurance, monthly expenditure, and health facility type (i.e., where the participant was employed).

Vaccine acceptance profile: The vaccine acceptance profile covers statements regarding willingness to accept the vaccine, attitudes towards COVID-19 vaccines, and motivation for getting vaccinated. In addition, willingness to receive the COVID-19 vaccine and factors that might potentially influence their decision-making process and vaccination recommendation behavior were assessed. Willingness to receive the vaccine was assessed using a binary scale (Yes and No). Multiple factors that could affect decision-making were assessed through Likert-scale questions and binary questions. This includes participants’ concerns regarding the safety and efficacy of COVID-19 vaccines and whether such concerns impacted their decisions regarding vaccines and vaccination. Responses were recorded on a Likert scale of 1 = strongly agree; 2 = agree; 3 = neutral; 4 = disagree; 5 = strongly disagree, and then converted into a three-point scale (1—Positive (Agree and Strongly Agree; 2—Neutral (Neither Agree nor Disagree); 3—Negative (Disagree and Strongly Disagree)) for multivariate logistic regression analysis.

Moreover, study participants were also asked about their willingness to pay for vaccines as well as whether they would recommend the vaccines to their family and patients.

Knowledge and perception profile: This set of questions assessed participants’ knowledge and perceptions around COVID-19, COVID-19 vaccines, and vaccination.

### 2.2. Data Collection and Study Sites

Data collection started after obtaining ethical clearance and piloting the questionnaire. Data collection was carried out in 4 provinces at 4 different sites, namely Hasan Sadikin General Hospital (Bandung, West Java), Syiah Kuala Hospital (Banda Aceh, Aceh), Mataram Province Hospital (Mataram, West Nusa Tenggara), and Dr. M. Haulussy Regional General Hospital (Ambon, Maluku).

The questionnaire was tested with 40 healthcare workers from Universitas Padjadjaran to evaluate its clarity. The main study was carried out in Bahasa Indonesia. A minimum sample size of 700 was determined based on the conservative assumption of a 50% acceptability rate, a 2.5% margin of error, and a 95% confidence interval. Participation in the study was entirely voluntary, and no compensation was provided to the participants.

The data collection was conducted after the questionnaire was developed and entered in SurveyMonkey© (San Mateo, CA, USA). The online questionnaire was then distributed by the site managers to the hospitals, adhering to the predetermined inclusion criteria. The data collection followed the first wave of COVID-19 vaccine deployment from January to March 2021 in Indonesia. The statement of informed consent was provided on the first page of the online questionnaire, to be completed before answering the questionnaire. All responses were recorded anonymously on the Survey Monkey platform and exported to an Excel file for data analysis.

### 2.3. Variable Definition and Statistical Analysis

The variables were categorized into two groups: categorical and continuous. Categorical variables were analyzed using Fisher’s exact test, while continuous variables were assessed using Student’s *t*-test. Logistic regression, employed for both univariate and multivariate analyses, aimed to determine predictors of vaccine acceptance among healthcare workers. The outcome was defined as the willingness to receive the COVID-19 vaccine upon its availability. Initially, a univariate logistic regression analysis was executed to assess predictors of COVID-19 vaccination acceptance. Variables demonstrating a *p*-value < 0.20 in the univariate analysis were subsequently included in a multivariate logistic regression model, wherein regression coefficient (B), Wald Chi-square, *p*-value, odds ratios (ORs), and corresponding 95% confidence intervals (95%CIs) were computed. A significance level of 0.05 was set, and a lower *p*-value indicated significance. Additionally, Friedman’s test was applied to determine the mean rank for sources of information on the COVID-19 vaccine and vaccination. Data analysis was performed using IBM^®^ SPSS^®^ Statistical Software, Version 29.0.

### 2.4. Ethical Approval

The study was approved by the Universitas Padjadjaran Research Ethics Committee (REF_1137/UN6/KEP/EC/2020).

Note: The methodology used in this study is similar to the one used in another study (https://www.mdpi.com/2076-393X/11/3/683, accessed on 15 April 2024). However, it is important to note that the study population is different. This study concentrates on healthcare workers, whereas the other focuses on university students and lecturers.

## 3. Results

### 3.1. Socio-Demographic Characteristics

A total of 3112 responses were recorded on the Survey Monkey platform (Version 2.0.5); 380 responses were excluded from the analysis due to an incomplete response, leaving 2732 healthcare workers included in the final analysis. The total number varied under different variable categories, as the study incorporated the number of samples based on the completion of data collection for each variable.

Table 1 illustrates the current demographic of participants targeted in the research study. The majority, comprising 60.5%, fell within the age range of 26–35 years, while 23.7% were aged between 36–45 years. Most respondents were doctors (55%) in their early careers, including general practitioners, medical residents, and university staff. Most of them also work for the university hospitals (74.0%), followed by satellite facilities affiliated with the university, such as primary care facilities (16.0%) and/or special COVID-19 facilities (2.9%). Due to their association with university hospitals, most participants held at least a master’s or specialist degree or doctoral (PhD) degree. This is reflected in Table 1, where the majority of the participants’ highest education was bachelor’s/MD degree (63%) followed by a diploma (18.6%) (a diploma is awarded after completing professional education that focuses on mastering certain applied skills) and a master’s degree (10.7%). Most participants were from West Java province (41.9%), followed by Aceh (28.0%), West Nusa Tenggara (16.9%), and Maluku (13.2%). Above 60% of the participants were female, and nearly three-quarters (74.2%) of the respondents were married.

### 3.2. Willingness to Receive the COVID-19 Vaccine and Trust in the Vaccination Programs

The majority (80.39%) of the healthcare workers were willing to receive the COVID-19 vaccine, while nearly one-fifth (19.61%) expressed hesitancy towards the vaccine. Around two-thirds of the respondents (63.3%) indicated that their trust in vaccination remained unaffected by the COVID-19 pandemic. This suggests that these participants maintained confidence in the health benefits of vaccines and trusted vaccination programs, even prior to the emergence of the pandemic. Conversely, 36.7% of the respondents reported a shift in their trust regarding COVID-19 vaccination (Figure 1).

In addition, the cost of COVID-19 vaccines seemed to influence HCWs’ decisions regarding their acceptance. Almost 42% of HCWs were open to paying for the COVID-19 vaccines based on its price, while 37.3% opted against it. Just over one-fifth of the respondents (20.96%) were willing to pay for COVID-19 vaccines (Figure 1).

### 3.3. Reasons for Not Being Willing to Receive the COVID-19 Vaccine

Among the participants who were not willing to receive the vaccine, over 40% expressed concerns about the vaccine’s safety, while 30.8% were anxious about potential side-effects following vaccination (Figure 2).

### 3.4. Socio-Demographic Factors Associated with the COVID-19 Vacciation Decision

A cross-tab analysis revealed that the acceptance rate of the COVID-19 vaccine was slightly lower in Aceh (60.7%) compared to the other three provinces, where the acceptance rate was 81–91%. Binary logistic regression analysis showed that the residents of West Java were more inclined to accept the COVID-19 vaccine (OR [95%CI]: 2.32 [1.69–3.18]) compared to those from West Nusa Tenggara. Conversely, participants from Aceh exhibited a lower inclination to receive the COVID-19 vaccine (OR [95%CI]: 0.34 [0.26–0.45]). Individuals with a diploma demonstrated lower acceptance rates for the COVID-19 vaccines compared to those with a medical degree (OR [95%CI]: 0.71 [0.56–0.89]), whereas respondents holding a master’s/specialty in medicine showed a greater willingness to receive the vaccine (OR [95%CI]: 1.67 [1.15–2.43]). Furthermore, participants working in primary care settings exhibited reduced willingness to receive the COVID-19 vaccine (OR [95%CI]: 0.50 [0.41–0.85]) compared to those in other healthcare facilities. Additionally, the acceptance of the COVID-19 vaccine was 1.7 times higher among male participants than female participants (OR [95%CI]: 1.70 [1.38–2.09]). Similarly, single participants were more inclined to accept the COVID-19 vaccine compared to married participants (OR [95%CI]: 1.29 [1.03–1.62]). Participants identifying as Hindu (OR [95%CI]: 5.69 [2.07–15.6]) and Christian (OR [95%CI]: 2.78 [1.97–3.93]) were more open to receiving the COVID-19 vaccine compared to those identifying as Muslim. However, no significant differences were observed between job title, age group, and type of health insurance in terms of COVID-19 vaccine acceptance (Table 2).

### 3.5. Attitude of Healthcare Workers towards the COVID-19 Vaccine and Vaccination

Figure 3 shows the preference of respondents when asked about COVID-19 vaccines and vaccination (Cronbach’s alpha (α)—0.805). Almost half (48%) of the respondents strongly believed that the COVID-19 vaccine was the most effective prevention method. Over one-third (37%) hold a neutral stance, and 15% expressed a negative sentiment, possibly questioning the effectiveness of vaccines compared to other methods. A strong positive sentiment is observed, with 69% expressing confidence in the vaccine’s ability to protect themselves and their families. In addition, over two-thirds (67%) of the respondents were opposed to the statement that the COVID-19 vaccine offers no benefit to them. The majority of the participants (63%) did not believe in their natural immunity and expressed a positive sentiment towards the necessity of the vaccine (Figure 3).

With regard to motivation to get vaccinated, the majority (52%) were positively influenced by their colleagues. Furthermore, positive family and friends’ influence to get vaccinated against COVID-19 was evident in 37% and 34% of the respondents, respectively. Around 40% of the participants believed that natural prevention methods were more effective than vaccines (Figure 3).

A significant proportion of the participants (55%) expressed concerns about serious side-effects that might be associated with the COVID-19 vaccine. Vaccine costs were also a significant concern, with 44% expressing neutrality and 23% expressing concerns about high costs. Safety concerns were prevalent among the participants, with 44% worried about the safety of the COVID-19 vaccine. Less than one-tenth of the participants (8%) stated that they would refuse the COVID-19 vaccine due to religious reasons, and 65% disagreed with rejecting the COVID-19 vaccine on religious grounds (Figure 3).

### 3.6. Other Factors Associated with the COVID-19 Vaccination Decision

The majority of the participants (2353/2685; 87.6%) had contacts with suspected COVID-19 patients at their workplace, yet only above one tenth of the respondents (370/2698; 13.7%) confirmed being infected with SARS-CoV-2. Moreover, over 60% of the healthcare workers in the study (1699/2692; 63.1%) were involved in treating COVID-19 patients. A significant majority of respondents (2132/2555; 83.4%) knew about the ongoing phase III COVID-19 vaccine clinical trial in Indonesia. The uptake of influenza vaccine among the healthcare participants in Indonesia appeared to be low, with only around one-fifth (522/2679; 19.5%) of the study participants having received a flu vaccine in the past five years (Table 3).

Multivariate logistic regression analysis showed that participants who were infected with SARS-CoV-2 (OR [95%CI]: 2.68 [2.11–3.40]), those with chronic disease (OR [95%CI]: 1.85 [1.46–2.34]), participants who were aware of the phase-III clinical trial in Indonesia (OR [95%CI]: 3.07 [2.44–3.87]), and those who had received the Flu vaccine within the past 5 years (OR [95%CI]: 1.92 [1.45–2.55]) demonstrated a positive attitude towards the COVID-19 vaccine (Table 3). In addition, healthcare workers’ recommendation behavior was positively associated with their willingness to accept the vaccine. Healthcare workers who would recommend the COVID-19 vaccine to their patients (OR [95%CI]: 17.67 [13.92–22.44]) and to their family members and friends (OR [95%CI]: 36.40 [27.97–47.38]) were more likely to accept the COVID-19 vaccine themselves (Table 3).

Furthermore, within the subset of participants who expressed readiness to receive the COVID-19 vaccine, factors such as the vaccine’s efficacy level, dosage regimen, and willingness to cover associated expenses significantly influenced their decision-making process. Notably, 64% (1717 out of 2677) indicated a preference for a vaccine with an efficacy of at least 80%, and only 21.8% of the respondents (584 out of 2677) were open to vaccines with efficacy ranging from 50% to 80%. Results from multiple regression analysis revealed a stronger inclination towards accepting a COVID-19 vaccine with an efficacy of ≥80% compared to those who cited alternative considerations regarding vaccine efficacy. Furthermore, concerning the financial aspect, participants willing to contribute financially towards the vaccine were 7.52 times more likely to accept it compared to those advocating for government-provided free vaccination services (OR [95%CI]: 7.52 [5.25–10.78]). (Note: the hypothetical estimated price ranged from IDR 200,000 to IDR 500,000 (USD 13.19 to USD 32.97)) (Table 4).

Moreover, a multivariate regression analysis was conducted using Likert scale statements, where the five-point scale was converted into a three-point scale (positive, neutral, and negative). This was done to assess how participants’ opinions influenced their willingness to receive the COVID-19 vaccines. Participants who believed that their daily activities would be disrupted if they contracted the virus (OR [95%CI]: 1.72 [1.06–2.78]), those who considered COVID-19 a serious illness (OR [95%CI]: 2.71 [1.47–4.97]), individuals who had confidence in the effectiveness of the COVID-19 vaccine for prevention (OR [95%CI]: 9.82 [7.23–13.33]), or those who had high trust in the government for providing transparent and timely information (OR [95%CI]: 9.46 [7.05–12.69]) were more inclined to accept the COVID-19 vaccine compared to those expressing negative sentiments toward these statements (Table 5). Conversely, participants who were apprehensive about the safety of the COVID-19 vaccine (OR [95%CI]: 0.06 [0.04–0.10]) and potential post-vaccination side effects (OR [95%CI]: 0.10 [0.06–0.18]), those with strong religious convictions (OR [95%CI]: 0.025 [0.018–0.036]), and individuals who believed in natural methods for preventing COVID-19 (OR [95%CI]: 0.045 [0.032–0.064]) were less inclined to accept the COVID-19 vaccine (Table 5).

### 3.7. Sources of Information on COVID-19, COVID-19 Vaccines and Vaccination, and Suitable Platforms for Communication

The channels individuals use to access information can significantly shape their perceptions and decisions regarding vaccines. In our study, it was evident that social media platforms (such as Facebook, WhatsApp, Instagram, and Twitter) were the primary sources of information on COVID-19, COVID-19 vaccines, and vaccination. Following social media, radio, newspapers, and television were identified as the second most common sources of information. Scientific journals ranked third in frequency as a source of information. Conversely, the Ministry of Health and COVID-19 Task Force website, along with fellow colleagues, were cited as the least common sources of information by participants in the study (Table 6).

The participants were also asked about the most suitable media for disseminating information on COVID-19 vaccines and vaccination in Indonesia. The majority of the participants, i.e., 65% and 64.3%, highlighted direct counselling, outreach and promotion, and social media (Facebook, Instagram, Twitter) as the most suitable platforms. Television and the Ministry of Health website were listed as the third (58.2%) and fourth (43.6%) most suitable media for communication. Posters/leaflets/brochures and radio were considered the least preferable media for communicating information on COVID-19 vaccines and vaccination (Figure 4).

### 3.8. Information to Be Communicated for COVID-19 Vaccine Introduction in Indonesia

When asked about the types of information that needs to be disseminated during the early phase of vaccine introduction in Indonesia, the majority of the participants emphasized the effectiveness (69.5%) and importance of the COVID-19 vaccine (68.2%), followed by information on the risks and safety profile of the COVID-19 vaccine. Information on COVID-19 prevention (42.5%) and how the virus is transmitted (40.1%) were listed as the least priority information to be communicated for vaccine introduction in Indonesia (Figure 5).

## 4. Discussion

The main objective of this research study was to assess the level of COVID-19 vaccine acceptance among healthcare workers in Indonesia and identify the factors influencing vaccine acceptance or hesitancy. In our study, it was seen that above 80% of the healthcare workers were willing to receive the COVID-19 vaccine. This shows that the COVID-19 vaccine acceptance among Indonesian healthcare workers is lower compared to similar studies conducted in Singapore (94.9%) [27], the United Kingdom (93.4%) [28], and Lebanon (86.8%) [29]. On the contrary, healthcare workers in Indonesia were more willing to accept the COVID-19 vaccine compared to the healthcare workers surveyed in the United States (36%) [30], a joint study conducted in France, Belgium, and Canada (72.4%) [31], a review of 35 studies conducted globally in 2020–2021 (average vaccine acceptance rate of 77.49%) [32], in Oman (42.3%) [22], Zambia (72.1%) [33], the Democratic Republic of the Congo (DRC) (67.2%) [34], and Somalia (61%) [35]. The varying rates of COVID-19 vaccine acceptance illustrate that the vaccine acceptance rate is highly contextual in terms of place and time, varying from one nation to the next and from one time point to the next, is influenced by numerous factors, and is a true reflection of trust and confidence in vaccines, the system delivering them, sources of information on these topics, and in the government and scientific communities.

In our study, the vaccine acceptability rate was lower among certain socio-demographic characteristics, including marital status, where single participants were more willing to receive the COVID-19 vaccine, which is contrary to the findings in the DRC, where married participants were more willing to receive the vaccine [34]. The exact reason for this discrepancy is unclear but could likely be influenced by the cultural and lifestyle differences between the countries. In addition, males were more likely to accept the vaccines in this study. Similar findings were reported in the US [30], Lebanon [29], and Zambia [33]. Some studies have outlined a higher risk for COVID-19-related complications and deaths among men, which may have increased risk perception among the male participants, leading to a higher willingness to be vaccinated against COVID-19 [36]. Some socio-demographic factors like age and job title (physician, nurse) were significant predictors of vaccine acceptance in other studies [29,35], unlike in our study. Therefore, identification of socio-demographic factors can play a crucial role in designing and implementing targeted interventions to boost vaccine uptake among specific groups/sub-groups.

Participants with a lower education level (diploma) were less likely to accept the vaccine compared to those with a medical degree. This illustrates that participants with a higher medical education are more knowledgeable about vaccines and vaccination through their education or medical practice. The COVID-19 vaccine acceptance rate among healthcare workers in primary healthcare settings like private practice, clinics, and puskesmas (government authorized community health posts across Indonesia) was lower than among those in secondary care settings like hospitals. This could be likely due to fewer patients seeking primary healthcare services once infected with SARS-CoV-2. Additionally, during the early phase of the COVID-19 pandemic, Indonesia restructured its healthcare services, designating some hospitals exclusively for COVID-19 patients [37].

Religion and place of residence were significant predictors of willingness to receive the COVID-19 vaccine in our study. Participants following Hinduism and Christianity were more likely to accept the vaccine compared to those practicing Islam, possibly because Hindu and Christian participants were less concerned about the vaccine’s halal status. Nevertheless, some COVID-19 vaccines authorized in Indonesia were halal-certified to boost vaccination rates among the Muslim population [38]. In addition, participants from Aceh had a lower vaccine acceptance rate compared to those from West Java and West Nusa Tenggara. Similar findings were reported by Machmud P.B. et al. and Khatiwada M. et al. to understand hepatitis B vaccination and COVID-19 vaccine acceptance, respectively, in Aceh, where participants from Aceh were less willing to receive the hepatitis B vaccine as well as the COVID-19 vaccine [39,40]. The likely reason for this is that Aceh is one of the provinces in Indonesia with a majority Muslim population, which might have an impact on vaccine acceptance. A recent report highlighted that Aceh has one of the lowest vaccination coverages in Indonesia, with only 42.7% of children aged 12–23 months completing their basic vaccination schedule [41]. Vaccine hesitancy among healthcare workers can negatively impact the overall immunization program, since the general public relies on them to seek advice on vaccines and vaccination.

Acceptance of the COVID-19 vaccine was also linked to a willingness to cover its cost. Harapan et al.’s study [42] found that more than three-quarters of participants (78.3%; 1065 out of 1359) were prepared to pay for and receive the COVID-19 vaccine. However, in Indonesia, COVID-19 vaccines were provided free of charge, with healthcare workers prioritized for vaccination, mirroring practices in many countries worldwide [43]. Furthermore, given the global and national urgency of the COVID-19 pandemic, healthcare workers recognized the necessity of widespread access to vaccination, funded by the government. In this study, 64.1% of participants expressed a willingness to receive a COVID-19 vaccine only if it had an efficacy of ≥80%, and only 9.9% indicated they would accept the vaccine regardless of its efficacy. These findings suggest that vaccine efficacy is one of the important factors influencing healthcare workers’ decisions to get vaccinated against COVID-19 in Indonesia. Similar results were previously reported in Indonesia, where 93.3% of respondents preferred a COVID-19 vaccine with 95% effectiveness [44].

In our study, the perceived risk of COVID-19 infection emerged as a key predictor of vaccine acceptance, similar to findings from studies in the United States [30] and the Democratic Republic of the Congo [34]. Participants who viewed COVID-19 as a severe threat to themselves and their families were more likely to accept vaccination. Hence, it is imperative to raise awareness about the severe complications associated with COVID-19. Additionally, having experienced SARS-CoV-2 infection was correlated with a greater likelihood of accepting the COVID-19 vaccine, as direct exposure to the disease positively influences one’s perception of its risk. Our study also revealed that healthcare workers who had previously received influenza vaccination were more inclined to accept the COVID-19 vaccine, consistent with findings from studies in the United States and Italy [30,45]. Furthermore, healthcare workers who were aware of the Phase III clinical trial of the COVID-19 vaccine in Indonesia were more likely to accept it, suggesting that early engagement with vaccine information can enhance understanding and positively impact vaccination decisions. Moreover, participants who trusted the government for information on COVID-19 vaccine development and rollout were more likely to accept the vaccine. Therefore, it is vital for the government to provide transparent and regular updates on the vaccine and address public concerns promptly to maintain trust in vaccination programs and the healthcare system administering them.

Healthcare workers are seen as the most trusted source of information on vaccines and play a crucial role in fostering trust between the public and immunization programs [46]. An initial reluctance among healthcare workers to receive the COVID-19 vaccine can have significant negative impacts. Studies have shown that healthcare workers who are willing to get vaccinated are more likely to recommend vaccines to their friends, family, and patients [47,48]. Our study also demonstrated this strong association, as healthcare workers who intended to get vaccinated were more likely to recommend the COVID-19 vaccine to their family members and patients.

In our study, participants cited the effectiveness of COVID-19 vaccines in preventing the disease and contributing to the containment of the pandemic as a key motivation for vaccination. Conversely, the main reasons for vaccine hesitancy were concerns about the safety of COVID-19 vaccines and potential side effects. These safety concerns are global, as highlighted by studies in the US [30], a review of 35 international studies (conducted from 2020–2021) [32], and research studies in Europe [48]. Therefore, it is of utmost importance to disseminate information on vaccine safety, efficacy, and vaccination guidelines on a regular basis and organize training programs for healthcare workers on vaccination programs and ways to differentiate trustworthy information from misinformation and disinformation. Furthermore, our study demonstrated that some healthcare workers changed their trust in vaccination due to the pandemic, which might influence acceptance of the COVID-19 vaccine. However, many factors are at play during pandemic times, such as lack of information as well as misinformation and disinformation, among other factors. At the point when information about COVID-19 vaccines was limited, such as at the early start of the pandemic, healthcare workers often delayed the decision around vaccination. Conversely, misinformation leads to individuals taking measures based on fear, which conflicts with facts or evidence [49]. In HCWs, misinformation also puts an emotional burden on them, such as distress and confusion, as false information contradicts their medical knowledge and perspectives [49]. Therefore, disseminating accurate information through medical agencies and professional societies is vital to boosting confidence and vaccine uptake among healthcare workers.

Despite social media being the main source of information for participants on COVID-19, the COVID-19 vaccine, and vaccination, they emphasized that direct counseling, outreach, and promotion programs are the most effective ways to communicate this information to the public. This preference likely stems from the fact that, while social media is a convenient platform, the abundance of information, including misinformation, available on these platforms can negatively impact people’s vaccination decisions. Participants emphasized the need to clearly communicate the importance, safety profile, and potential side-effects of the COVID-19 vaccine to enhance confidence and maintain trust in vaccination programs.

This study opens doors to deeper investigation regarding vaccine uptake behaviors among HCWs in Indonesia, especially in emergency situations like a pandemic. As was seen in the detailed literature search, there are only limited studies looking at aspects of vaccine uptake and recommendation among HCWs in Indonesia. Important questions that can be further investigated include the degree of association between the level of knowledge and vaccine-recommending behaviors of HCWs, the effects of sociocultural forces like religion on vaccination behavior among HCWs, and the reasons why there is a disparity in vaccine acceptance among different professions (physicians, nurses, pharmacists, etc.). Tailored information campaigns, along with education and training programs for different healthcare professions, will play a vital role in helping people understand and address their concerns about vaccines and vaccination, thereby positively influencing the vaccination decisions of the general public in the long term. A previous study conducted in Indonesia demonstrated lower COVID-19 vaccine acceptance among students and lecturers [40]. Therefore, training HCWs can also promote vaccination among the public since the study reflected healthcare workers as one of the most trusted sources of information on vaccines and vaccination. It is also equally important to understand the role of healthcare workers in delivering vaccination programs in Indonesian settings to tailor the vaccine education and training programs. Information from this study will also help to identify the root cause of vaccine hesitancy among HCWs, isolate the weak points of the COVID-19 vaccination campaign drive of the government, and create better crafted education programs that will be more targeted to the needs and concerns of HCWs. In addition, it is still relevant to look at the past, and if we encounter similar pandemic situations like COVID-19 (hopefully not), the findings of this study can be useful to develop tailored information campaigns as well as design education and training programs for healthcare workers for the timely and effective introduction and implementation of vaccination programs.

One notable drawback of the study was its reliance on online methods, potentially excluding healthcare workers without access to smartphones, email, or the internet. Furthermore, the study’s scope was restricted to four sites across four provinces, limiting its applicability to other regions in Indonesia. Moreover, because all data were collected online, the accuracy of responses, including those related to influenza vaccination history, could not be verified. The cross-sectional nature of the survey prevented the establishment of causality between attitudinal factors and intentions. Additionally, since data collection occurred at a single time point, changes in attitudes towards vaccination following the availability of COVID-19 vaccines could not be captured. Conducting surveys at multiple time points might have provided deeper insights or identified trends in vaccine acceptance.

## 5. Conclusions

Our study found a moderate-to-high level of acceptance of the COVID-19 vaccine among healthcare workers in four Indonesian provinces, with notably lower acceptance rates in Aceh compared to the other three provinces. Factors such as perceptions of disease risk, history of flu vaccination, vaccine efficacy, and trust in the government were significant predictors of COVID-19 vaccine acceptance among healthcare workers in Indonesia. Major concerns about the safety profile of the COVID-19 vaccine and potential side effects following vaccination were cited as the primary reasons for vaccine hesitancy. Thus, it is crucial to provide timely and transparent information on COVID-19 vaccines and vaccination to address these concerns and overcome barriers to the implementation of vaccination programs in Indonesia.

These findings were briefed to the Ministry of Health, Indonesia, in June 2021, which played a vital role in realigning the COVID-19 vaccination implementation strategies and developing targeted interventions to increase vaccination uptake among healthcare workers.

## Figures and Tables

**Figure 1 vaccines-12-00654-f001:**
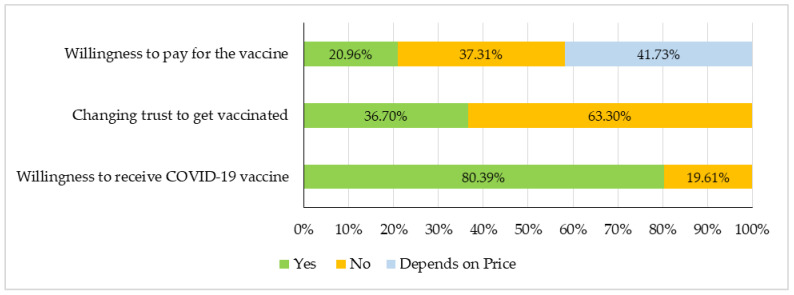
HCWs response to willingness to receive vaccination, change in trust in vaccination, and willingness to pay for the vaccine. Total number of responses: 2732.

**Figure 2 vaccines-12-00654-f002:**
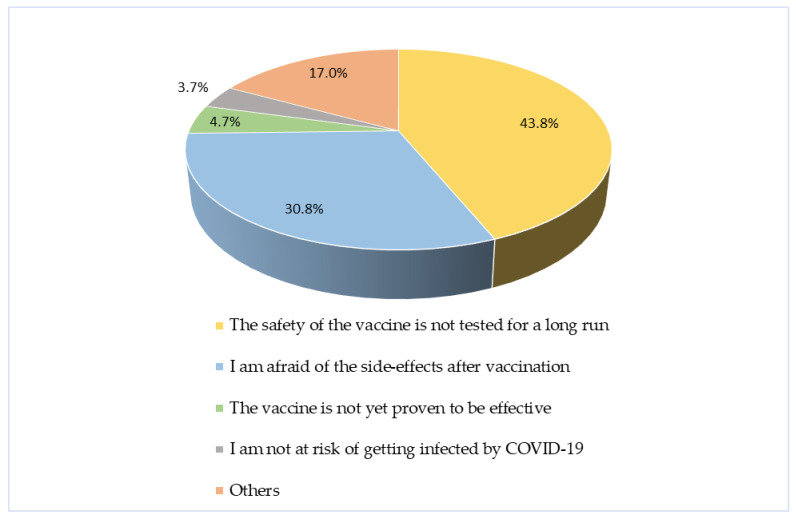
Pie chart representing various reasons for not being willing to accept the COVID-19 vaccine among healthcare workers in Indonesia. Total number of responses: 493.

**Figure 3 vaccines-12-00654-f003:**
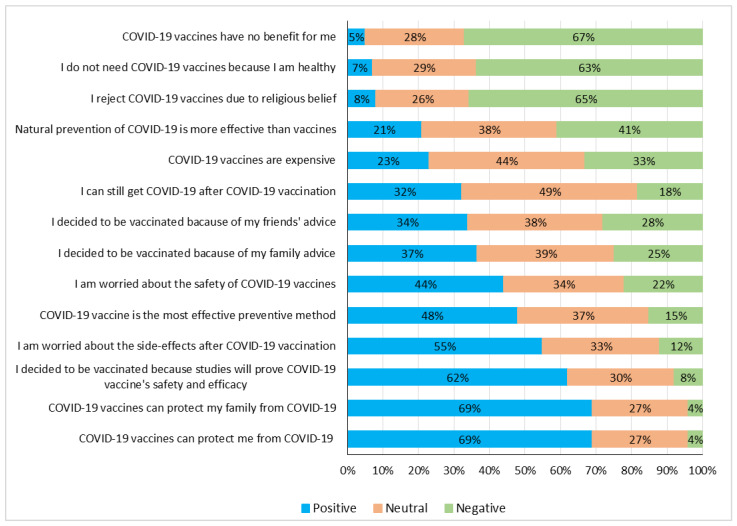
Bar graph showing HCWs’ level of agreement on the COVID-19 vaccine and vaccination. Total number of responses: 2699.

**Figure 4 vaccines-12-00654-f004:**
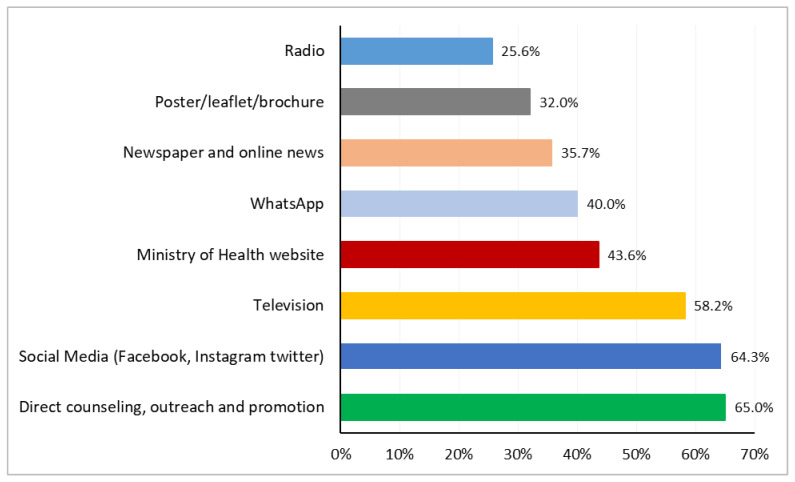
Bar graph showing the suitable platform for communication on COVID-19 vaccines and vaccination. Total number of responses: 2732.

**Figure 5 vaccines-12-00654-f005:**
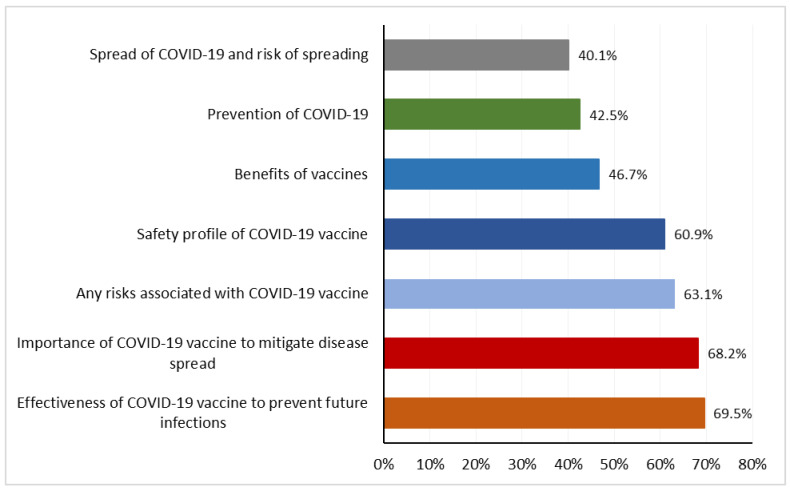
Bar graph showing the participants views on the information to be communicated for the introduction of COVID-19 vaccine. Total number of responses: 2732.

**Table 1 vaccines-12-00654-t001:** Socio-demographic characteristics with frequency (n) and percentages (%).

Socio-Demographic Variables	Frequency (n)	Percentages (%)
Province (University)		
Jawa Barat (West Java)	1140	41.9
Aceh	764	28.0
Nusa Tenggara Barat (West Nusa Tenggara)	460	16.9
Maluku	360	13.2
Educational Level		
Bachelors/MD degree	1702	63.0
Diploma	502	18.6
Doctorate	48	1.8
Masters	289	10.7
Sub-Specialists	47	1.7
Others	112	4.1
Job Title		
Physician	1498	55.0
Nurse/Nursing staff	896	32.9
Pharmacist	27	1.0
Midwives	94	3.4
Others (clinical psychologist, medical technician, public health staff, nutritionist, physical therapists)	211	7.7
Healthcare facility type		
Primary care (private practice, clinics, primary health care (puskesmas, posbindu, emergency facilities), treatment center, delivery home)	2016	74.0
Secondary care (hospitals including emergency hospitals, both public-owned and private)	434	16.0
Others	271	10.0
Age		
18–25	158	5.8
26–35	1637	60.5
36–45	643	23.7
46–55	196	7.2
56–65	74	2.7
Gender		
Male	1063	39.3
Female	1643	60.7
Marital Status		
Single	697	25.8
Married	2008	74.2
Religion		
Islam	2184	80.7
Hinduism	85	3.1
Christianity	424	15.7
Buddhism	12	0.4
Others	2	0.1
Average monthly household expenditure (Indonesian Rupiah)		
˂1 million	64	2.4
1- < 5 million	1105	40.9
5- < 10 million	916	33.9
10- < 20 million	284	10.5
≥20 million	90	3.3
Prefer not to say	242	9.0
Type of health insurance you/your family have		
National health insurance	2240	82.4
Personal/private insurance	77	2.8
Both insurance	272	10.0
No insurance	128	4.8

**Table 2 vaccines-12-00654-t002:** Binary logistic regression analysis of willingness to receive COVID-19 vaccine with socio-demographic factors.

Socio-Demographic Characteristics	Willingness to Receive COVID-19 Vaccine Yes/No n (%)	Willingness to Receive COVID-19 Vaccine Yes vs. No		
		B	*p*-Value	OR (95 C.I.)
Province (N = 2695)				
Jawa Barat	1032 (91.3)/98 (8.7)	0.84	**<0.001 ****	2.32 (1.69–3.18)
Aceh	455 (60.7)/295 (39.3)	−1.07	**<0.001** **	0.34 (0.26–0.45)
Maluku	309 (86.8)/47 (13.2)	0.37	0.06	1.45 (0.98–2.14)
Nusa Tenggara Barat	376 (81.9)/83 (18.1)	Ref		
Educational Level (N = 2671)				
Bachelors/MD Degree	1367 (81.0)/320 (19.0)	Ref		
Diploma	371 (75.1)/123 (24.9)	−0.35	**0.004 ****	0.71 (0.56–0.89)
Doctorate (PhD)	43 (91.5)/4 (8.5)	0.92	0.08	2.52 (0.90–7.06)
Masters/Specialty	250 (87.7)/35 (12.3)	0.51	**0.007 ***	1.67 (1.15–2.43)
Others	83 (74.1)/29 (25.9)	−0.40	0.07	0.67 (0.43–1.04)
Sub-Specialists	38 (74.1)/8 (17.4)	0.11	0.79	1.11 (0.51–2.40)
Job Title (N = 2698)				
Physician	1263 (85.0)/223 (15.0)	0.43	0.10	1.53 (0.91–2.56)
Nurse/Nursing staff	640 (72.4)/244 (27.6)	−0.34	0.19	0.71 (0.42–1.19)
Pharmacist	20 (76.9)/6 (23.1)	−0.10	0.84	0.90 (0.32–2.54)
Others	175 (84.1)/33 (15.9)	0.36	0.25	1.43 (0.77–2.66)
Midwives	74 (78.7)/20 (21.3)	Ref		
Healthcare facility type (N = 2694)				
Primary care	1572 (78.6)/429 (21.4)	−0.52	**0.005 ***	0.59 (0.41–0.85)
Secondary care	368 (86.0)/60 (14.0)	−0.005	0.98	0.99 (0.64–1.55)
Others	228 (86.0)/37 (14.0)	Ref		
Age (N = 2678)				
18–25	128 (81.5)/29 (18.5)	−0.44	0.28	0.64 (0.29–1.44)
26–35	1317 (81.1)/306 (18.9)	−0.47	0.19	0.62 (0.31–1.27)
36–45	495 (78.0)/140 (22.0)	−0.67	0.07	0.51 (0.25–1.06)
46–55	155 (80.7)/37 (19.3)	−0.50	0.21	0.61 (0.28–1.33)
56–65	62 (87.3)/9 (12.7)	Ref		
Gender (N = 2678)				
Male	898 (85.4)/154 (14.6)	0.53	**<0.001 ****	1.70 (1.38–2.09)
Female	1259 (77.4)/367 (22.6)	Ref		
Marital status (N = 2676)				
Single	578 (83.4)/115 (16.6)	0.26	**0.03 ***	1.29 (1.03–1.62)
Married	1577 (79.5)/406 (20.5)	Ref		
Religion (N = 2678)				
Islam	1681 (77.9)/478 (22.1)	Ref		
Hinduism	80 (95.2)/4 (4.8)	1.282	**<0.001 ****	5.69 (2.07–15.60)
Christianity	382 (90.7)/39 (9.3)	1.02	**<0.001 ****	2.78 (1.97–3.93)
Others	14 (100.0)/0 (0.0)	-	-	-
Monthly household expenditure (N = 2673)				
Low	45 (72.6)/17 (27.4)	Ref		
Medium	1607 (80.3)/395 (19.7)	0.43	0.14	1.54 (0.87–2.71)
High	315 (84.7)/57 (15.3)	0.74	**0.02 ***	2.09 (1.12–3.90)
Prefer not to say	186 (78.5)/51 (21.5)	0.32	0.32	1.38 (0.73–2.61)
Type of health insurance (N = 2688)				
National health insurance	1757 (79.4)/455 (20.6)	−0.06	0.78	0.94 (0.60–1.47)
Private health insurance	68 (88.3)/9 (11.7)	0.61	0.15	1.83 (0.81–4.17)
Both insurance	238 (87.8)/33 (12.2)	0.56	0.054	1.75 (0.99–3.09)
No insurance	103 (80.5)/25 (19.5)	Ref		

Note: B: regression coefficient; *p*-value; OR.: odds ratio (non-adjusted); 95 C.I.: 95 confidence interval; Ref: reference category; *p*-value significant at <0.05; *p*-value < 0.05: *; *p*-value < 0.005: **; monthly household expenditure; low: <1 million IDR; medium: 1 million IDR- < 10 million IDR, and high: ≥10 million IDR; religion; others: Buddhism and others.

**Table 3 vaccines-12-00654-t003:** Multivariate logistic regression analysis of different factors with respect to (w.r.t.) willingness to get vaccinated.

Characteristics	Frequency	Willingness to Get Vaccinated		Willingness to Receive COVID-19 Vaccine: Yes vs. No		
		Yes n (%)	No n (%)	B	*p*-Value	aOR (95 C.I.)
Infected with SARS-CoV-2 (N = 2698)						
Yes	370	239 (64.6)	131 (35.4)	0.98	**<0.001 ****	2.68 (2.11–3.40)
No	2328	1933 (83.0)	395 (17.0)	Ref		
Contacts with suspected COVID-19 patients at work (N = 2685)						
Yes	2353	1898 (80.7)	455 (19.3)	−0.05	0.71	0.95 (0.71–1.26)
No	332	265 (79.8)	67 (20.2)	Ref		
Have chronic disease (N= 2663)						
Yes	442	317 (71.7)	125 (28.3)	0.62	**<0.001 ****	1.85 (1.46–2.34)
No	2221	1831 (82.4)	390 (17.6)	Ref		
Received Flu vaccine during the last 5 years (N = 2679)						
Yes	522	458 (87.7)	64 (12.3)	0.65	**<0.001 ****	1.92 (1.45–2.55)
No	2157	1700 (78.8)	457 (21.2)	Ref		
Treating COVID-19 patients (N = 2692)						
Yes	1699	1368 (80.5)	331 (19.5)	0.003	0.98	1.003 (0.82–1.22)
No	993	800 (80.6)	193 (19.4)	Ref		
Recommend your patients take the COVID-19 vaccine (N = 2552)						
Yes	2050	1881 (91.8)	169 (8.2)	2.87	**<0.001 ****	17.67 (13.92–22.44)
No	502	194 (38.6)	308 (61.4)	Ref		
Recommend your family members and friends take the COVID-19 vaccine (N = 2552)						
Yes	2015	1904 (94.5)	111 (5.5)	3.59	**<0.001 ****	36.40 (27.97–47.38)
No	537	172 (32.0)	365 (68.0)	Ref		
Belief towards vaccines and vaccination changed after the onset of the COVID-19 pandemic (N = 2550)						
Yes	935	779 (83.3)	156 (16.7)	0.21	0.051	1.23 (0.99–1.52)
No	1615	1295 (80.2)	320 (19.8)	Ref		
Awareness of phase III COVID-19 vaccine clinical trial in Indonesia (N = 2555)						
Yes	2132	1803 (84.6)	329 (15.4)	1.12	**<0.001 ****	3.07 (2.44–3.87)
No	423	271 (64.1)	152 (35.9)	Ref		

Note: N = total respondents; B: regression coefficient; aOR: odds ratio (adjusted); 95 C.I.: 95 confidence interval; Ref: reference category; *p*-value significant at <0.05; *p*-value < 0.005: **.

**Table 4 vaccines-12-00654-t004:** Multivariate logistic regression analysis of vaccine efficacy, number of doses, mode of administration, and willingness to pay as predictors of the acceptance of the COVID-19 vaccine.

Characteristics	Frequencyn (%)	Willingness to Be Vaccinated		Intent to Be Vaccinated: Yes vs. No		
		Yes, n (%)	No, n (%)	B	*p*-Value	aOR (95 C.I.)
Vaccine Efficacy (N = 2677)						
I would take the COVID-19 vaccine if it is ≥80% effective.	1717 (64.1)	1333 (77.6)	384 (22.4)	1.59	**0.001 ****	4.90 (3.30–7.28)
I would like to take the COVID-19 vaccine if it is even 50–80% effective.	584 (21.8)	551 (94.3)	33 (5.7)	3.16	**<0.001 ****	23.59 (14.09–39.52)
I would take the COVID-19 vaccine irrespective of the vaccine efficacy data.	265 (9.9)	228 (86.0)	37 (14.0)	3.66	**<0.001 ****	8.70 (5.21–14.55)
Others	111 (4.2)	46 (41.4)	65 (58.6)	Ref		
Number of doses (N = 2650)						
I would consider the number of doses as one of the criteria to receive COVID-19 vaccine.	1957 (73.8)	1634 (83.5)	323 (16.5)	0.65	**<0.001 ****	1.92 (1.57–2.36)
I would not consider the number of doses as one of the criteria to receive COVID-19 vaccine.	693 (26.2)	502 (72.4)	191 (27.6)	Ref		
Willingness to pay for the COVID-19 vaccine (estimated price: IDR 200,000–IDR 500,000) (N = 2551)						
Yes	537 (21.1)	500 (93.1)	37 (6.9)	2.02	**<0.001 ****	7.52 (5.25–10.78)
Depends on the actual vaccine price.	1069 (41.9)	964 (90.2)	105 (9.8)	1.63	**<0.001 ****	5.11 (4.02–6.51)
No	945 (37.0)	607 (64.2)	338 (35.8)	Ref		

N: total number; B: regression coefficient; aOR: odds ratio(adjusted); 95 C.I.: 95 confidence interval; Ref: reference category; *p*-value significant at <0.05; *p*-value < 0.005: **.

**Table 5 vaccines-12-00654-t005:** Multivariate logistic regression analysis of willingness to receive the COVID-19 vaccine with positive responses for Likert-scale statements.

Statements	Intent to Be Vaccinated: Yes vs. No		
	B	*p*-Value	aOR (95 C.I.)
All my activities will be disrupted if I get infected with SARS-CoV-2.	0.54	**0.02 ***	1.72 (1.06–2.78)
COVID-19 is a serious disease for me.	0.99	**0.001 ****	2.71 (1.47–4.97)
COVID-19 is a serious disease for my family.	1.42	**<0.001 ****	4.14 (1.87–9.12)
The COVID-19 vaccine is the most effective tool to prevent COVID-19.	2.28	**<0.001 ****	9.82 (7.23–13.33)
I decided to be vaccinated because my workplace suggested so.	1.92	**<0.001 ****	6.81 (5.07–9.16)
I decided to be vaccinated because a close friend contracted COVID-19.	2.13	**<0.001 ****	8.39 (5.88–11.96)
I decided to be vaccinated because studies will prove the long-term safety and efficacy of COVID-19 vaccination.	3.07	**<0.001 ****	21.53 (15.42–30.13)
One of the ways to prevent COVID-19 complications is through COVID-19 vaccination.	3.31	**<0.001 ****	27.52 (19.00–39.84)
I do not like injections in general.	−1.10	**<0.001 ****	0.33 (0.26–0.42)
I am worried about the safety of the COVID-19 vaccine.	−2.76	**<0.001 ****	0.06 (0.04–0.10)
I do not need the COVID-19 vaccine because I am healthy.	−3.78	**<0.001 ****	0.023 (0.016–0.033)
I am worried there will be side effects after COVID-19 vaccination.	−2.28	**<0.001 ****	0.10 (0.06–0.18)
I prefer not to get vaccinated because of my religious beliefs.	−3.68	**<0.001 ****	0.025 (0.018–0.036)
I believe that natural COVID-19 disease prevention is better than vaccination.	−3.10	**<0.001 ****	0.045 (0.032–0.064)
I might still get COVID-19 even when I am vaccinated with the COVID-19 vaccine.	−1.43	**<0.001 ****	0.24 (0.16–0.34)
My government provides transparent and up-to-date information on COVID-19 vaccine development and its introduction.	2.25	**<0.001 ****	9.46 (7.05–12.69)
I value the importance of vaccines and vaccination more now, after the onset of the COVID-19 pandemic.	2.79	**<0.001 ****	16.33 (10.54–25.30)
I valued the importance of vaccines and vaccination before the onset of the COVID-19 pandemic as well.	2.27	**<0.001 ****	9.66 (6.47–14.45)

Note: The statements were recorded on a 5 point scale (Strongly Agree, Agree, Neither Agree nor Disagree, Disagree, Strongly Disagree) and converted into a 3 point scale (Positive: Strongly Agree and Agree, Neutral: Neither Agree nor Disagree, Negative: Disagree and Strongly Disagree). The above multivariate regression analysis is shown only for positive responses with negative response as the reference category. B: regression coefficient; aOR: odds ratio (adjusted); 95 C.I.: 95 confidence interval; *p*-value significant at <0.05; *p*-value < 0.05: *; *p*-value < 0.005: **.

**Table 6 vaccines-12-00654-t006:** Friedman’s Test mean score on sources of information regarding COVID-19, the COVID-19 vaccine, and vaccination.

Information Sources	Mean Score
The rank of the main sources of information	
Social media (Facebook, WhatsApp, Instagram, Twitter)	2.27
Radio, Newspaper, Television	3.00
Scientific Journals	3.48
Workplace	3.59
Ministry of Health and COVID-19 Task Force Website	4.07
Fellow Colleagues	4.59

Note: The lowest mean score in the non–parametric test (Friedman’s test) analysis implies the first rank.

## Data Availability

The data presented in this study are available upon request from Universitas Padjadjaran. The data are not publicly available for reasons of privacy.

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
