# Peer review of "Understanding COVID-19 Vaccine Acceptance among Healthcare Workers in Indonesia: Lessons from Multi-Site Survey"

_vaccines, 2024, doi:10.3390/vaccines12060654_

Round 1

Reviewer 1 Report (Previous Reviewer 2)

Comments and Suggestions for Authors

The revisions are fine.  The explanation that this study is relevant because it is unique in Indonesia is fine if acceptable to the editors.  The data would be more useful if they were comparable to some other countries in the area.  But, the study is fine as it stands.  

Author Response

Dear Reviewer,

Thank you very much for your thoughtful feedback and for recognizing the revisions made.

We are glad to hear that you find the explanation of our study’s relevance acceptable. We acknowledge your suggestion regarding the comparability of our data with other countries in the region, and we will explore ways to address this in future research.

We truly appreciate your time and consideration.

Reviewer 2 Report (Previous Reviewer 4)

Comments and Suggestions for Authors

The authors have addressed reviewer comments, not extensively but adequately. 

Author Response

Dear Reviewer,

Thank you for you feedback on the revisions made to our manuscript. We appreciate your acknowledgment that we have adequately addressed the reviewer comments, albeit not extensively. Your assessment is valuable to us, and we have taken your feedback into consideration.

In response to your observation, we have made additional revisions to the manuscript to further enhance its clarity, coherence, and overall quality. Throughout the manuscript, the changes made are marked in red (for added text) and strikethrough (for removed text).

Your input has been instrumental in guiding the revision of our manuscript, and we are grateful for your continued engagement with our manuscript.

Thank you once again for your time and valuable feedback.

This manuscript is a resubmission of an earlier submission. The following is a list of the peer review reports and author responses from that submission.

Round 1

Reviewer 1 Report

Comments and Suggestions for Authors

Abstract:

Lines 23-24: Add an “a” before “priority group.”

Line 25: Consider changing “aimed” to “was conducted.”

Line 28: Add a “the” before a “COVID-19 Vaccine.”

Line 29: Add a “the” before “safety profile” and change “COVID-19 Vaccine” to “COVID-19 Vaccines.”

Line 31-32: Consider changing this statement “Being male, single, having higher education level and higher risk perception would increase the acceptability of COVID-19 vaccine…” to the following: “Male gender, single status, higher level of education, and perception of the COVID-19 risks of infection increased the acceptability of the COVID-19 vaccine.”

Line 32-33:  Consider revising the following statement, “High trust in the government and more confidence on vaccine safety and efficacy studies would motivate the participants to receive COVID-19 vaccine…” as follows: “Other motivators of COVID-19 acceptance include a high level of trust in the government and increased confidence in the validity of the vaccine safety and efficacy trials.”

Line 34: Remove “fact-based” and change “on timely” to “in a timely” manner. Consider revising the remaining part of the sentence because it is unclear. Did the authors mean that the HCWs COVID-19 educational programs are crucial for increasing HCWs’ confidence in the benefits of the COVID-19 vaccine?  Please clarify and revise this statement as appropriate.

Introduction:

Lines 39-40: Remove “human” prior to “lives” and change “across the globe” to “globally” to increase conciseness.

Line 41:  For the phrase, “health systems globally,” remove “globally to avoid redundancy.

Line 43 and through the manuscript:  Please a “the” prior to “WHO.”

Line 98: Please consider changing the tense of the statement so that it conveys the potential benefits of this research study.

Study design, development of questionnaire

The authors should include the exclusion criteria.

Line 105: Remove “, the criteria of which entail” because it is redundant and change to “if they were…” and remove “aged.”

Line 105: Throughout the text, be consistent with the tense. For example, change “working” to “worked.”

Lines 107-9:  The sentence beginning with “The study protocol and questionnaire…” was unclear. Please clarify and revise. Why did the authors include an “early national study”? Was the protocol and questionnaire based upon methods conducted in the national study?  The authors should consider removing “reflecting the status of COVID-19 at the time of research” since it seems extraneous.

Social demographic profile

Line 116:  Research studies cannot “also assessed.”  Please revise. Consider starting out the sentence with “Social demographic include….”

Vaccine profile:

This section should be revised to enhance conciseness and clarity. What did the authors mean by the COVID-19 vaccine profile.  Were study participants asked questions about their concerns regarding the safety and effectiveness of the COVID-19 vaccines?  The authors need to explain how the “set of questions were adopted and adjusted.”

Data Collection and sites:

Line 146: What did the authors mean by “and the questionnaire required no modification” ? Please clarify and revise the statement or delete.

Line 147: Why did the authors include the “national language of Indonesia”?  Consider deleting or clarifying and revising this statement.

Variable definition and statistical analysis

Line 164:  Consider changing to “seek” to “identify.”

Line 165:  Remove “particularly” from the phrase “particularly using logistic regression.”

Lines 166-169: Please revise this statement beginning with “The variables included…” to enhance clarity and conciseness.

Results:

Line 190-191: “2,732 healthcare workers included in the final analysis.”  However, in Table 2, the numbers are different (e.g. Province N-2695, Job Title N-2698). Please verify results and revise if necessary.

Lines 191-3: Please revise this section to increase clarity and conciseness.

Line 197: Please further delineate “specialist trainees.”  Did the authors mean that these trainees were medical residents or fellows specializing in specific areas of medicine or surgery>

Line 197: Consider changing “university staffs” to “university faculty” or “university staff.”

Line 198: What did the authors mean by “which is the secondary hospital”? Please revise to enhance clarity or delete.  Consider changing “Most of them” to “Most participants.”

Lines 203-4: What did the authors mean by “followed by diploma” and “masters or specialist”? Please define “specialist.” Please revise to enhance clarity.

 Line 213 Consider changing “in getting vaccination” to “in getting vaccinated.”

Lines 214-15: Consider changing “that vaccines provide health benefits and trusted vaccination programs and were ready to accept vaccination services” to “that vaccines provided health benefits, trusted vaccination, programs and were to accept vaccination services….”

Line 215 Add a “the” before the “COVID-19 pandemic.”

Lines 216-218: Revise this section to enhance conciseness.

Lines 219-221: Change “price” to “cost.”

Line 222:  Change “just above one-fifth” to “over one-fifth” to increase conciseness and change “COVID-19 vaccine” to “COVID-19 vaccines.”

Line 242: Add a “the” before the COVID-19 vaccine.

Line 243: Please be specific and concise with the description of these respondents’ level of education: “Participants who had diploma as their highest education level.” Did the authors mean respondents with a PhD?

Line 243, 246: Please lower case “Participants.”

Line 253-256:  Change “COVID-19 vaccine” to “the COVID-19 vaccine.”

Line 273: Change “Above one-third” to “Over one-third.”

Line 277: Change “opposed” to “were opposed.”

Line 280-282: Consider making this section more concise.

Line 284- 285: Change “colleague’s influence” to “their colleagues’ influence on their decision to receive a COVID-19 vaccination.”

Line 285-286: Change “friend influence” to “friends’ influence.”

Line 287-289: Consider making this section more concise.

Line 293-295: Change “vaccine cost was” to “vaccine costs were….” and change “high cost” to high costs.”

Line 297: Change “below one-tenth” to “less than one-tenth.”

Line 301: Add a “The” prior to “Majority.”

Line 312: Change “had received Flu vaccine” to “had received the Flu vaccine.”

Line 315:  Change “wiliness” to “willingness.”

Line 329-330.  Consider changing “that they would take the vaccine that is proven to be ≥ 80%” to “that they would take the vaccine that is proven to be at least 80% effective.”      

Line 332: Consider changing “with ≥ 80%” to “with ≥ 80% effectiveness.”

Line 334: Consider changing “cost for COVID-19 vaccine” to “costs for Covid-19 vaccines.”

Line 346: Consider changing “neural” to “neutral.”

Line 347-348: Consider changing “COVID-19 vaccine” to “the COVID-19 vaccines.”

Line 407: Consider changing “majority of the participants” to “the majority of the participants.”

Line 408-409: Consider changing “information on risk associated with COVID-19 vaccine and safety profile of the COVID-19 vaccine” to “information on risks and the safety profile associated with the COVID-19 vaccine.”

Lines 410-411: Consider changing “the ways COVID-19 spread (40.1%) were listed as least priority” to “the ways that COVID-19 is disseminated (40.1%) were listed as the least priority….”

Discussion:

Lines 420-425: Please revise this section to enhance clarity and conciseness.

 Line 433: Define DRC.

Line 450: Define “puskesmas.”

 Line 497: Change “deposed” to “likely.”

 Lines 539-542:  This sentence was unclear and lengthy. Please revise.

Table 1

Define BPJS

Table 3

Consider changing “Treating COVID-19 patient” to “Treating COVID-19 patients.”

Consider changing “Recommend your patients to take COVID-19 vaccine “Recommend your patients take the COVID-19 vaccine”.

Consider changing “Recommend your family members and friends to take COVID-19 vaccine” to “Recommend your family members and friends take the COVID-19 vaccine”.

Consider changing “Belief towards vaccine and vaccination changed after the onset of COVID-19 pandemic” to “Belief towards vaccine and vaccination changed after the onset of the COVID-19 pandemic”.

Comments on the Quality of English Language

Please edit the manuscript to increase conciseness, flow, and clarity. See above comments. Thank you very much.

Reviewer 2 Report

Comments and Suggestions for Authors

I am not sure why this article is relevant now.  The data are three years old and focus on vaccine acceptance rates just after they became available.  This study might have been useful three years ago, but not relevant today.  The pandemic is over and why should we care about these issues now, and particularly in four areas in Indonesia?  Therefore, I would recommend rejecting the manuscript or asking the authors to explain why the study is relevant today.  Should the authors collect the same data now and compare the results to see if vaccines are more acceptable given the outcome of the pandemic?  Just not sure why this study is needed.   

Reviewer 3 Report

Comments and Suggestions for Authors

This paper highlights the various factors influencing COVID-19 vaccine attitudes among health care workers in Indonesia. It does a thorough job of assessing many different factors and presents some interesting trends. One of my main concerns, however, is timeliness. These data were collected between December 2020 and February 2021. I wonder at how relevant it is to today, given the rapidly changing access to and information about the COVID vaccine. Perhaps it is still relevant to look at the past, but you certainly can’t make comparative statements such as was made on lines 418 and 419 of the Discussion where acceptance rates were compared to Singapore, UK, and Lebanon when those studies were not necessarily conducted at the same time.

The statistics are thorough, but I do have some questions and suggestions. First, I feel that sufficient detail is lacking for the validity and reliability of the survey. The authors suggested that the pilot tested the survey to a handful of health care workers and determined that no changes were needed, but I don’t feel that is sufficient. I would suggest additional reliability measures (e.g., confirmatory factor analysis, cronbach’s alph, etc.) and additional validity measures (e.g., expert validity, concurrent validity, etc.). We need to feel a little more confident that the survey was indeed measuring what the authors intended.

Additionally, I was confused as to why the Likert scale items were collapsed into 3-point scales, rather than maintaining the full data at a 5-point scale, especially since the authors used multivariate regression – to treat a Likert item as scalar, you would want to maintain the 5-point scale. Can the authors spend a little more time statistically justifying the decision to collapse these data?

Lastly, I just have a few suggestions for clarity:

Line 203, the use of “diploma” is not internationally known – can you define this for a more generalized audience?

I was confused by the reference to respondents changing their trust in COVID-19 vaccination with the pandemic (see line 216, for example). It was unclear as to whether they changed for the worse or for the better. Can you clarify?

In Figure 2, it occurred to me that there was likely overlap in these categories (i.e., there may be some respondents who worried about safety AND were afraid of side-effects). Were respondents forced to choose only one response? Perhaps a different graphic would display this overlap better.

In Figure 3, this graph might be more impactful if you ordered the questions from most agreed with at the top to least agreed with at the bottom.

On line 313, where the authors are listing factors affecting COVID-19 attitudes, it should say that those factors influence attitudes, rather than saying that these factors “were willing to receive”.

In all of your regression tables, you should include the standardized beta so that the reader can easily interpret the slope of the relationship.

On line 372, the authors claim that the sources through which people got their information played a “vital role in shaping their perception…”. But, these data just assessed where they got the information, not what influence it had on their opinions. I would remove that claim.

Line 516 – COVID-10 should be COVID-19.

And, incidentally, there are grammatical mistakes throughout the paper. The authors would benefit from using an English editing service.

Overall, it is an interesting paper and I look forward to reading a revision.

Comments on the Quality of English Language

Extensive grammatical errors throughout.

Reviewer 4 Report

Comments and Suggestions for Authors

The authors have submitted data on health care professionals' vaccination perceptions during the early days of the COVID-19 pandemic. That period of time is receding in time, of course, but nonetheless readers might be interested in this work because of the specific focus on health care professionals. 

To optimize the utility of the piece for readers, the authors should consider expanding their analysis of sources of information. That was a somewhat unique angle to the paper and yet the authors just provide general descriptive information. They should consider looking at whether people in different roles tend to use different channels as that would have implications for the proactive communication effort the authors recommend at the end.

The authors also should connect this paper to literature on sources of information among health care professionals. For example, they could cite a paper such as: M’ikanatha, N. M., Lautenbach, E., Kunselman, A. R., Julian, K. G., Southwell, B. G., Allswede, M., Rankin, J. T., & Aber, R. C. (2003). Sources of bioterrorism information among emergency physicians during the 2001 anthrax outbreak. Biosecurity and Bioterrorism, 1(4), 259-265.

There were at least two predictions in the paper which were compelling and which deserve more comment. First, socioeconomic status, e.g., income, seems to predict vaccination willingness and so the authors should consider whether that might reflect structural resource differences. Second, the predictive value of previous flu vaccination is worth noting. That result connects with other literature and so the authors ought to connect this discussion to such literature, e.g., Southwell, B. G., Kelly, B. J., Bann, C. M., Squiers, L. B., Ray, S. E., & McCormack, L. A. (2020). Mental models of infectious diseases and public understanding of COVID-19 prevention. Health Communication, 35(14), 1707-1710.

At the end of the paper, the authors also should more clearly note the implications of their work for future campaigns to reach health care professionals. Without that sort of discussion, the manuscript is too descriptive and insufficient as an explicit theoretical contribution.